# Demography and Fitness of *Anastatus japonicus* Reared from *Antheraea pernyi* as a Biological Control Agent of *Caligula japonica*

**DOI:** 10.3390/insects13040349

**Published:** 2022-03-31

**Authors:** Xiao-Yan Wei, Yong-Ming Chen, Xingeng Wang, Rui-E Lv, Lian-Sheng Zang

**Affiliations:** 1Institute of Biological Control, Jilin Agricultural University, Changchun 130118, China; xyw11282022@126.com; 2United States Department of Agriculture, Agricultural Research Service, Beneficial Insects Introduction Research Unit, Newark, DE 19713, USA; xingeng.wang@usda.gov; 3Institute of Walnut, Longnan Economic Forest Research Institute, Longnan 746000, China; LRE390952@163.com; 4Key Laboratory of Green Pesticide and Agricultural Bioengineering, Guizhou University, Guiyang 550025, China

**Keywords:** biological control, *Anastatus japonicus*, *Caligula japonica*, *Antheraea pernyi*, two-sex life table, mass rearing

## Abstract

**Simple Summary:**

Japanese giant silkworm (JGS), *Caligula japonica* Moore, is an emerging defoliator pest of forest and fruit trees in East Asia, causing severe economic losses. In this study, we used eggs of the Chinese oak silkworm (COS), *Antheraea pernyi* Guérin-Méneville, as an alternative host to rear the dominant JGS egg parasitoid, *Anastatus japonicus* Ashmead. We showed that *A. japonicus* could be efficiently reared on COS eggs. Furthermore, *A. japonicus* reared from COS eggs had a high biological control potential against JGS. This study provides useful information on the demography, parasitism rate, and rearing methods of *A. japonicus* on COS eggs and the basis for developing a cost-effective biological control program against JGS using *A. japonicus*.

**Abstract:**

Japanese giant silkworm (JGS), *Caligula japonica* Moore, is an emerging defoliator pest of forest and fruit trees in East Asia, causing severe economic losses. To develop a cost-effective biological control program against JGS, we used eggs of the Chinese oak silkworm (COS) *Antheraea pernyi* Guérin-Méneville as an alternative host to rear the most dominant JGS egg parasitoid *Anastatus japonicus* Ashmead. We compared the demographic parameters and total parasitism (killing) rates of *A. japonicus* parasitizing JGS and COS eggs using an age-stage, two-sex life table method. The results showed that *A. japonicus* performed differently on these two different hosts. *Anastatus japonicus* reared from COS eggs had a higher fecundity (369.7 eggs per female) and a longer oviposition period (35.9 days) on the COS than JGS eggs (180.9 eggs; 24.0 days). Consequently, *A. japonicus* parasitizing COS eggs had a higher intrinsic rate of increase (*r* = 0.1466 d^−1^), finite rate of increase (λ = 1.1579 d^−1^) and net reproductive rate (*R*_0_ = 284.9 offspring) than those parasitizing JGS eggs (*r* = 0.1419 d^−1^, λ = 1.1525 d^−1^, *R*_0_ = 150.0 offspring). The total net parasitism rate (the number of parasitized hosts in which the parasitoids successfully developed) of *A. japonicus* parasitizing COS eggs was 284.9, significantly higher than that of *A. japonicus* parasitizing JGS eggs (150.0), while the net non-effective parasitism rate (the number of parasitized hosts in which the parasitoids failed to develop) of the former (0.0) was significantly lower than that of the latter (9.6). These results suggest that *A. japonicus* can be efficiently reared on the alternative (or factitious) COS eggs, and the reared parasitoids have a high biological control potential against the target JGS.

## 1. Introduction

Native to East Asia, the Japanese giant silkworm (JGS) *Caligula japonica* Moore is widely distributed in China, Japan, North Korea, and The Russian Far East [1]. In China, 38 plant species belonging to 30 genera in 20 families have been reported as hosts of this pest; the main hosts are walnut (*Juglans regia* L.) [2,3], ginkgo (*Ginkgo biloba* L.) [4], and chestnut (*Castanea mollissima* Blume) [5]. Since it was first found unexpectedly in 2006 in Kangxian and Huixian Counties in Longnan City, Gansu Province, China, JGS has rapidly spread and outbroke in an area of 240 hectares, becoming an emerging pest in this region [5]. Current management strategies for this pest rely on repeated applications of chemical pesticides [6,7], which have led to resistance of this pest to some chemical pesticides [8]. Extensive use of insecticides also poses a serious threat to the environment, natural enemies, and ecological balance [9]. Therefore, it is crucial to develop effective and environmentally friendly alternatives for the sustainable management of this emerging pest. Biological control, especially by means of self-perpetuating and effective parasitoids, could be a viable option for the sustainable and eco-friendly management of insect pests [10,11].

Egg parasitoids have been used as major biological control agents for the control of some invasive agricultural and forestry pests. For example, *Anastatus bifasciatus* Geoffroy has been reported as a predominant egg parasitoid specie against the invasive brown marmorated stink bug *Halyomorpha halys* Stål in Europe [12,13,14,15]. *Anastatus acherontiae* Narayanan, Subba Rao and Ramachandra (later identified as *Anastatus echidna* Motschulsky), and *Anastatus bangalorensis* Mani and Kurian were two important parasitoids of the litchi stink bug *Tessaratoma javanica* Thunberg in India [16,17]. *Anastatus fulloi* Sheng and Wang was also a notable example of successful biological control of *Tessaratoma papillosa* Drury in litchi orchards in China [18,19]. Previously, Chen et al. [20] reported that *A. japonicus* was the most dominant JGS egg parasitoid during field surveys in Gansu Province, China. They found that this egg parasitoid was able to parasitize different ages of JGS eggs [20]. The parasitoid could also be successfully reared on eggs of the Chinese oak silkworm (COS) *Antheraea pernyi* Guérin-Méneville, and the COS egg seemed to be an ideal factitious host for mass rearing of this parasitoid because of its simple operation method, low production cost, and convenient transportation [21,22]. However, it is still unclear how *A. japonicus* reared from an alternative host (COS egg) would perform on the target host (JGS egg), i.e., the potential of mass-reared *A. japonicus* from this non-natural host for the biological control of JGS. Therefore, this study aimed to comparatively evaluate the reproductive performance of *A. japonicus* reared from COS eggs on these two different hosts.

A life table is an indispensable tool for population ecology and pest management to quantify population demographic parameters such as survival, development, longevity, and fecundity [23,24]. Compared to the traditional female age-specific life table, a two-sex life table method integrates both age-stage and two-sex components to accurately estimate important life-history traits such as age-stage or age-specific survival for both sexes and age-specific fecundity for females and generates important demographic parameters [23,24]. This approach is, therefore, a valuable tool for comparative analyses of a parasitoid’s life-history strategies under different conditions and has been widely used in the analysis of life table of insects such as *Altica cyanea* Weber, *Trichogramma achaeae* Nagaraja, and Nagarkatti, *Aphelinus asychis* Walker, and *Encarsia formosa* Gahan [25,26,27,28,29,30,31,32,33]. Here, we compared and quantified population demographic parameters and total parasitism rates of *A. japonicus* parasitizing COS and JGS eggs using the computer programs of TWOSEX-MSChart [34] and CONSUME-MSChart [35] based on the age-stage, two-sex life table method [23,24,36]. The study provides the basis for mass rearing of *A. japonicus* on COS eggs for an effective biological control program of JGS.

## 2. Materials and Methods

### 2.1. Host Insects

Studies were conducted at the Institute of Biological Control, Jinlin Agricultural University, China. Both the Japanese giant silkworm (JGS), *C. japonica,* and the Chinese oak silkworm (COS) *A. pernyi* were collected from fields. COS cocoons were collected during late fall every year from fields in Yongji City, Jilin Province, China, and held at 4 °C in a refrigerator (dark) until late February, when the cocoons were transferred to an emergence room (25 ± 1 °C, 60% ± 10% RH and natural photoperiod) for adult emergence. Unfertilized COS eggs were extracted from mature and virgin female moths by squeezing the abdomen of the moths, washed with distilled water, and then air-dried at room temperature prior to their use for the rearing or experiments [37].

JGS cocoons were collected in walnut orchards during the middle of June in Kangxian, Gansu Province, China. Approximately 300 cocoons were collected every year and placed in an insect cage (30 × 30 × 30 cm) in an open insectary so that they were exposed to similar field conditions and expected to emerge as field populations did from August to October. Newly emerged adults were paired and kept in insect rearing cages (50 × 50 × 50 cm). The adults were monitored daily for oviposition, and freshly laid eggs (<2 days) were collected for the experiments.

### 2.2. Parasitoid

A laboratory colony of *A. japonicus* was established with adult wasps that emerged from field-collected and parasitized JGS egg masses in walnut orchards in 2017 in Kangxian, Gansu Province (105–106° E, 32.9–33.7° N), China [1,38]. The parasitoid species was identified based on the morphological descriptions by Dr. Gary Gibson (Agriculture and Agri-Food Canada, Canadian National Collection of Insects) and was further confirmed by COI sequence (GenBank accession number: MK604240) [1,39]. The parasitoid was maintained on COS eggs (unfertilized, extracted and washed eggs, see Section 2.1) under controlled conditions (25 ± 1 °C, 70% ± 5% RH, and 14 L:10 D photoperiod) [37] and had been reared for six generations on COS eggs before they were used in this study. Briefly, newly emerged wasps (<6 h) were collected into cylindrical and transparent plastic containers (14.0 × 9.0 cm, height × diameter) covered with fine mesh, with 30% honey water provided as food. Ten pairs of 3-day-old wasps were provided with ≈300 eggs in each container. After a 24 h exposure, the wasps were removed, and the exposed hosts were monitored until the emergence of wasps.

### 2.3. Performance of A. japonicus Reared from COS Eggs on COS and JGS Eggs

All experiments were conducted under laboratory conditions (25 ± 1 °C, 70% ± 5% RH, and 14 L:10 D photoperiod) and used the parasitoids reared from COS eggs as described above. To determine the age-specific survival and fecundity of *A. japonicus* on COS eggs, a pair of newly emerged (<6 h) female and male wasps was first mated in a glass tube (12 × 3 cm, length × diameter) and then provided with 40 COS eggs (unfertilized, extracted, and washed eggs, see Section 2.1) and 30% honey water as food in the tube. The host eggs were glued on a strip egg card (6 × 1.5 cm, length × width). New COS eggs and honey water were replaced daily until the death of the female wasp. If the male wasp died before the female, it was replaced with another male. For each replicate, the exposed egg card from the previous day was transferred to a new glass tube (7 × 2 cm, length × diameter). The sex and date of each emerged wasp were recorded daily until the wasp emergence had ceased for 20 consecutive days. Finally, all host eggs without emergence holes were dissected to determine the presence or absence of recognizable parasitoid cadavers under a stereomicroscope (Leica DFC 450, M165C, Singapore). There were 20 valid replicates (5 replicates were excluded because the female wasps either escaped during the replacement of fresh egg cards or were accidentally killed). Male and female *A**. japonicus* adults that emerged on the same day were paired and used to initiate the cohort in this study, which might not reflect the natural emergence pattern of the parasitoid population. Therefore, a bootstrap-match technique based on the life table method as described by Amir-Maafi et al. was used in this study [40].

Similarly, for the test of JGS eggs, a pair of newly emerged (<6 h) wasps reared from COS eggs was mated in a glass tube (12 × 3 cm, length × diameter), with 40 newly JGS eggs and 30% honey water provided in the tube. New JGS eggs and honey water were also replaced daily, and the survival of both female and male wasps was monitored until the death of the female. Any male who died before the female was also replaced with another male. There were 24 valid replicates (1 replicate was excluded because the female wasp escaped during the replacement of fresh egg cards). The rest of the experimental procedures were similar to those described above for the test of COS eggs.

### 2.4. Data Analysis

Data were analyzed according to the two-sex life table method [23,24,36]. The lifetime longevity, fecundity (i.e., the number of *A. japonicus* offspring successfully emerged from parasitized hosts), and non-effective (reproductive) parasitism rate (i.e., the number of *A. japonicus* offspring failed to develop or emerge from parasitized hosts) and other life table parameters were estimated for each population parasitizing COS or JGS eggs.

The age-stage survival rate (*s_xj_*) describes the probability that a newly laid egg will survive to age *x* and developmental stage *j* [23] and is calculated as:(1)$$s_{xj}=\frac{n_{xj}}{n_{01}}$$
where *n*_01_ is the number of newborns at the beginning of the population and *n_xj_* is the number of individuals among *n*_01_ surviving to age *x* and stage *j*. Variation in developmental rates among individuals may result in an overlap of survival at different stages.

The age-specific survival rate (*l_x_*) represents the survival rate from egg to age *x*, which ignores the stage differentiation and is the sum of *s_xj_* of all stages at age *x*, while the age-specific fecundity (*m_x_*) represents the average number of eggs laid for the individuals survived to age *x*. The *l_x_* and *m_x_* are calculated as follows:(2)$$l_{x}=\overset{\beta }{\underset{j=1}{\sum }}s_{xj}\;m_{x}=\frac{\sum_{j=1}^{\beta }s_{xj}f_{xj}}{\sum_{j=1}^{\beta }s_{xj}}$$
where *β* is the number of stages and *f_xj_* refers to the effective parasitism rate (i.e., the number of parasitoids successfully emerged from parasitized hosts) [41]. The age-specific net reproductive rate of population (*l_x_m_x_*) is the product of *l_x_* multiplying *m_x_*, which represents the average number of successfully parasitized eggs by all surviving individuals in the population at age *x*.

The life expectancy (*e_xj_*) represents the remaining survival time of individuals at age *x* and stage *j* and can be calculated according to Chi and Su [42] as:(3)$$e_{xj}=\overset{\infty }{\underset{i=x}{\sum }}\overset{\beta }{\underset{y=j}{\sum }}s_{iy}^{\prime }$$
where $s_{iy}^{\prime }$ is the probability that an individual of *n_xj_* will survive to age *i* and stage *y*.

The net reproductive rate (*R*_0_) defines the average number of emerged offspring by a parasitoid during its lifetime and is calculated as:(4)$$R_{0}=\overset{\infty }{\underset{x=0}{\sum }}l_{x}m_{x}$$

The intrinsic rate of increase (*r*) refers to the average daily growth rate of the population when it reaches the stable age-stage distribution, which is calculated by the iterative bisection method and the Euler-Lotka equation with the age indexed from 0 [43]:(5)$$\overset{\infty }{\underset{x=0}{\sum }}e^{-r\left(x+1\right)}l_{x}m_{x}=1$$

The finite rate of increase (*λ*) refers to the average daily growth rate of the population when it reaches the stable age-stage distribution and is calculated as follows:(6)$$\lambda =e^{r}$$

The mean generation time (*T*) refers to the time required to achieve *R*_0_ when the population reaches a steady growth rate (*λ* and *r*) and is calculated as:(7)$$T=\frac{\ln R_{0}}{r}$$

The age-stage reproductive value (*v_xj_*) refers to the average contribution of individuals at age *x* and stage *j* to future population growth [44,45] and is calculated as:(8)$$v_{xj}=\frac{e^{r\left(x+1\right)}}{s_{xj}}\overset{\infty }{\underset{i=x}{\sum }}e^{-r\left(i+1\right)}\overset{\beta }{\underset{y=j}{\sum }}s_{iy}^{\prime }f_{iy}$$

The net non-effective parasitism rate *G*_0_ refers to the total number of host eggs that are killed due to parasitism without the successful emergence of parasitoid adults. It is calculated as:(9)$$G_{0}=\overset{\infty }{\underset{x=0}{\sum }}\overset{\beta }{\underset{j=1}{\sum }}s_{xj}g_{xj}$$

The age-stage total parasitism rate *p_xj_* is the sum of the effective parasitism rate (*f_xj_*) and the non-effective parasitism rate (*g_xj_*). Therefore, the net parasitism rate *P*_0_ is the sum of the net reproductive rate (*R*_0_) and is calculated as:(10)$$P_{0}=\overset{\infty }{\underset{x=0}{\sum }}\overset{\beta }{\underset{j=1}{\sum }}s_{xj}\left(f_{xj}+g_{xj}\right)=R_{0}+G_{0}$$

The transformation rate (*Q_p_*) is defined as the total number of hosts killed to produce a single parasitoid offspring and is calculated as:(11)$$Q_{p}=\frac{P_{0}}{R_{0}}=\frac{R_{0}+G_{0}}{R_{0}}$$

The TWOSEX-MSChart program was used to calculate these demographic parameters [34], while the CONSUME-MSChart computer program was used to analyze the non-effective parasitism rate [35]. The bootstrap technique with 100,000 resamplings was used to estimate the standard errors of these population parameters [34]. Paired bootstrap test in the TWOSEX-MSChart program was used to test the significance of differences in the parameters (*p* ≤ 0.05) [46]. All figures were created using Sigma Plot 14.0.

## 3. Results

### 3.1. Longevity and Lifetime Fecundity of A. japonicus Parasitizing COS and JGS Eggs

As shown in Table 1, the initial cohort sizes of *A. japonicus* were 20:20 (female:male) on COS eggs and 24:24 (female:male) on JGS eggs, and the matched cohort sizes of the parasitoid were 259:77 (female:male) on COS eggs and 175:36 (female:male) on JGS eggs. The female adult longevity, female total longevity, male adult longevity, and male total longevity of *A. japonicus* parasitizing COS eggs were significantly longer than those parasitizing JGS eggs. However, no significant differences were observed in the adult preoviposition period and total preoviposition period between the parasitoids parasitizing COS and JGS eggs. The oviposition period and number of eggs laid per oviposition day by *A. japonicus* parasitizing COS eggs were higher than those parasitizing JGS eggs. The lifetime fecundity of the parasitoid was higher on the COS than JGS eggs. The age at *l_x_* ≤ 0.5 by the parasitoid parasitizing COS eggs was significantly delayed when compared to those parasitizing JGS eggs.

### 3.2. Age-Stage and Age-Specific Survival Rate and Fecundity of A. japonicus Parasitizing COS and JGS Eggs

The age-stage survival rate (*s_xj_*) revealed significant differences between *A. japonicus* parasitizing COS and JGS eggs (Figure 1). Adult females parasitizing COS eggs survived 79 days, which was longer than those parasitizing JGS eggs (52 days). However, the peak of the *s_xj_* at the adult stage of adult female parasitoids was higher on JGS eggs (82.94%) at the age of 37 days than on COS eggs (76.49%) at the age of 33 or 34 days. On the COS eggs, the *s_xj_* started to decrease at the age of 34 days but remained relatively stable from 45 to 52 days of age. On the JGS eggs, however, the *s_xj_* remained stable from 32 to 37 days of age and then decreased sharply after the age of 37 days. The peak of the age-stage survival rate of adult male *A. japonica* was also higher on the COS (22.92%) than JGS eggs (16.11%) at the age of 25 days. Overall, adult females had much longer longevity than adult males (Figure 1).

The age-specific survival rate (*l_x_*), age-specific fecundity (*f_x_*_2_), age-specific fecundity of population (*m_x_*), and age-specific net reproductive rate (*l_x_m_x_*) of *A. japonicus* parasitizing COS eggs and JGS eggs were illustrated on Figure 2. The *l_x_* was higher on the JGS than COS eggs from 36 to 47 days of age. The highest *m_x_* was 12.80 at the age of 42 days on the COS eggs or 10.54 at the age of 35 days on the JGS eggs. Both the *m_x_* and *l_x_m_x_* decreased gradually on COS and JGS eggs (after 42 and 35 days old, respectively), except that the *m_x_* increased again from 79 to 82 days of age and then decreased again on the COS eggs. The ages at which adult females reached 90% cumulative fecundity on COS eggs (256.45) and JGS eggs (135.03) were 61 and 47 days, respectively.

### 3.3. Life Expectancy and Reproductive Value of A. japonicus Parasitizing COS and JGS Eggs

The life expectancy values (*e_xj_*) of a newly laid *A. japonicus* egg on COS eggs and JGS eggs were 58.99 and 51.09 days, respectively (Figure 3). The highest *e_xj_* for the adult females on COS eggs (43.63 days) was higher than that on JGS eggs (33.49 days) at 23 and 22 days of age, respectively. The highest *e_xj_* for the adult males on COS eggs (11.27 days) was also higher than that on JGS eggs (7.66 days) at the age of 22 days.

The reproductive value of *A. japonicus* (*v_xj_*) first gradually increased with age; reaching the peak at the ages of 32 days (86.69 d^−1^) and 30 days (65.39 d^−1^) when parasitizing COS eggs and JGS eggs, respectively, and then gradually decreased with age (Figure 4). The *v_xj_* of adult females dramatically increased when they started oviposition (Figure 4). The reproductive value at the age of zero (*v*_01_) was equal to the finite rate of increase (*λ*). The reproductive values of newly laid eggs of *A. japonicus* parasitizing COS eggs and JGS eggs were 1.16 and 1.15, respectively.

### 3.4. Population Parameters of A. japonicus Parasitizing COS and JGS Eggs

The population demographic parameters of *A. japonicus* parasitizing COS and JGS eggs are shown in Table 2. The net reproduction rate (*R*_0_), intrinsic rate of increase (*r*), finite rate of increase (*λ*), and mean generation time (*T*) of *A. japonicus* parasitizing COS eggs were all significantly higher than those parasitizing JGS eggs. There were significant differences in the numbers of female and male offspring produced between parasitoids parasitizing COS eggs and JGS eggs. The net non-reproductive parasitism rate (*G*_0_) on COS eggs was zero. The total net parasitism rate (*P*_0_) (that was the sum of female and male offspring and the net non-reproductive parasitism) was higher on the COS than JGS eggs. The transformation rate (*Q_p_*) was significantly lower on the COS than JGS eggs.

## 4. Discussion

In this study, we compared the demographic parameters and parasitism rates of *A. japonicus* parasitizing COS and JGS eggs. We demonstrated that the parasitoid could be efficiently reared on the alternative (or factitious) host COS eggs, and reared parasitoids from the COS eggs had a high biological control potential against the target host JGS.

The two-sex life table analysis allows a thorough understanding of the demography and parasitic effectiveness of *A. japonicus* and comparisons of some important life-history traits and key demographic parameters of the parasitoid parasitizing these two different hosts. Overall, we showed that *A. japonicus* was more effective on COS than JGS eggs. Most of these key demographic parameters, such as adult longevity, female ovipositional period, lifetime fecundity, and net parasitism rate, were higher when parasitizing COS eggs than JGS eggs. This indicates that the parasitoid has a higher per capita reproductive potential on the COS than JGS eggs. Previously, the COS egg was also shown to be a suitable factious host for rearing other egg parasitoids such as *T. dendrolimi* Matsumura, *T. chilonis* Ishii [11], and *Aprostocetus brevipedicellus* Yang and Cao [47]. Here, we further confirm that *A. japonicus* can be efficiently reared on COS eggs.

The different demographics and parasitic effectiveness of *A. japonicus* on COS and JGS eggs may be caused by several factors. First, both host quality and host egg surface characteristics (e.g., host egg size and age, eggshell shape, and chorion thickness) could affect host acceptance and the resultant parasitism by an egg parasitoid [20,48,49,50,51]. In this study, we used naturally laid JGS eggs, which have a hard surface and thick eggshell [20]. In contrast, the COS eggs we used were extracted from virgin female moths, washed to remove the outer layer surface substrates, and the eggshell was still soft [37]. It could be harder to penetrate a naturally laid JGS egg than a treated COS egg by the ovipositor of female *A. japonicus*. There may involve higher costs for *A. japonicus* to parasitize JGS than COS eggs in terms of host handling time and energy investment per oviposition, thereby reducing the parasitoid’s longevity and lifetime fecundity when parasitizing the JGS eggs [50,52]. Indeed, previous studies found that extracted and washed COS eggs from virgin female moths largely facilitated ovipositor penetration by several egg parasitoids [38]. These studies also indicated that the hardness of the host eggshell negatively reduced a female parasitoid’s oviposition rate and longevity [50]. For example, the fecundity of *Trichogramma confusum* Viggiani on the eggs of *Corcyra cephalonica* Stainton (eggshell thickness of 4.2 um) and COS eggs (eggshell thickness of 48.0 µm) was 147 and 47, respectively [53]. Second, natal host species could affect a parasitoid’s performance on different host species. Since *A. japonicus* had been reared for six generations on COS eggs in the laboratory and all tested wasps were reared from COS eggs, it is likely that the parasitoid might have adapted to this natal host. Many studies showed that many parasitoids performed better in or on the host they previously developed due to preimaginal learning [54,55,56,57]. For example, the parasitism rate and longevity of *Aphidius microlophii* Pennacchio and Tremblay were significantly increased after the parasitoid was reared on the non-natural host of the pea aphid *Acyrthosiphon pisum* Harris for several generations [54].

Regardless of the host species, the overall patterns of these life-history traits (age-stage or age-specific survival rate, age-specific fecundity, life expectancy, reproductive values, Figure 1, Figure 2, Figure 3 and Figure 4) of *A. japonicus* were similar when parasitizing COS and JGS eggs; the differences were the peak values of these curves that were higher on the COS than JGS eggs. One only exception was that the *s_xj_* peak of female *A. japonicus* was slightly higher on the JGS than COS eggs (Figure 1). We also noticed that although a JGS egg (egg length ≈ 2.40 mm, width ≈ 1.49 mm) was smaller than a COS egg (egg length ≈ 3.21 mm, width ≈ 2.82 mm), the percentage of female *A. japonicus* offspring on the JGS eggs (79.08%) was slightly higher than that on the COS eggs (72.03%). The *s_xj_* refers to the probability of an egg surviving to age *x* and stage *j* and could be affected by host quality (i.e., host suitability). Offspring sex allocation decisions could also be affected by host quality, among various other factors (e.g., female age, host species, host density, female wasp density, superparasitism, and environmental conditions) [57]. A previous study reported that female *A. japonicus* adults were able to detect the size and estimate the quality of a host egg through receptors located on the ovipositor so that they were capable of adjusting sex allocation decisions based on host quality [58]. In general, arrhenotokous parasitoids allocate more female offspring to high-quality hosts and more male offspring to low-quality hosts [57]. Similar findings were reported for *A. japonicus* on six different host species; there was a significantly positive correlation between the size of the host egg and the percentage of female offspring of the parasitoid [59]. These results suggest that the natural host is more suitable than the alternative host for offspring survival, and JGS eggs may have a higher quality than COS eggs. In addition, the COS eggs used in this study were not fertilized. Several studies corroborated that egg parasitoids preferred fertilized over unfertilized eggs [58,60]. For example, Krugner found that the egg parasitoids *Gonatocerus*
*morrilli* Howard and *Telenomus coloradensis* Crawford preferred to parasitize fertilized than unfertilized eggs of *Homalodisca vitripennis* Germar [61]. Although a high survival rate and a high percentage of female offspring are favorable for effective mass rearing for introduction and augmentative releases of parasitoids, these differences are generally marginal for *A. japonicus* parasitizing COS and JGS eggs. It is worth of mentioning that the cumulative fecundity (*l_x_m_x_*) reached 90% at the ages of 38 and 24 days when parasitizing COS and JGS eggs, respectively (Figure 2). This suggests that female adults older than 38 days are no longer reproductive and should not be used for the rearing of this parasitoid on COS eggs under similar conditions as the current study. Such information could help improve mass-rearing techniques and laboratory bioassays or field release when selecting the suitable age of females for rearing or studies.

In conclusion, *A. japonicus* has a high biological control potential against JGS as it exhibits a long longevity and ovipositional period as well as a high fecundity on JGS eggs. As demonstrated in this study, *A. japonicus* reared from COS eggs are callable of killing about 160 JGS eggs per female lifetime under the tested conditions. Effective methods for the rearing of *A. japonicus* and other related egg parasitoids on COS eggs have been investigated previously. The current study further demonstrated that *A. japonicus* reared from COS eggs had a high killing capacity on the target JGS. Our results thus have important implications for mass rearing of *A. japonicus* on this factitious host for augmentative release against JGS. However, further studies may be needed to evaluate the efficiency of this parasitoid in the field and environmental conditions (e.g., temperature and humidity) that may affect its efficiency in the field.

## Figures and Tables

**Figure 1 insects-13-00349-f001:**
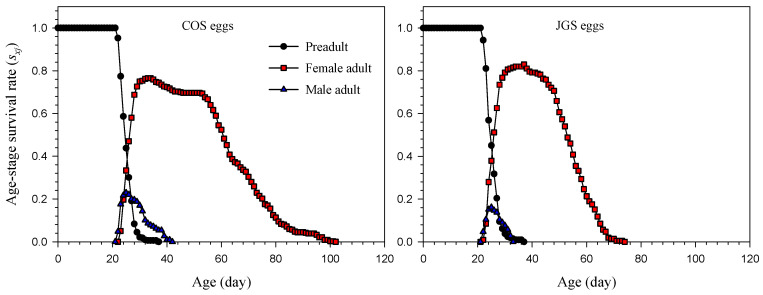
Age-stage survival rate (*s_xj_*) of *Anastatus japonicus* parasitizing COS and JGS eggs.

**Figure 2 insects-13-00349-f002:**
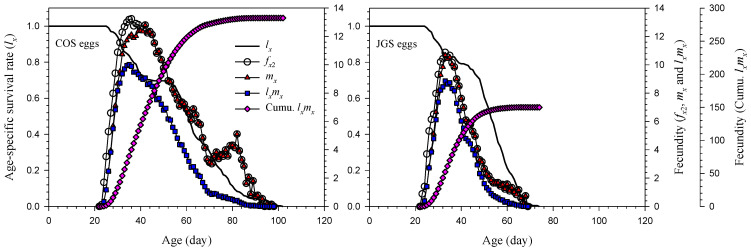
Age-specific survival rate (*l_x_*), age-specific fecundity (*f_x_*_2_), age-specific fecundity of population (*m_x_*), age-specific net reproductive rate of population (*l_x_m_x_*), and cumulative fecundity of *Anastatus japonicus* parasitizing COS and JGS eggs.

**Figure 3 insects-13-00349-f003:**
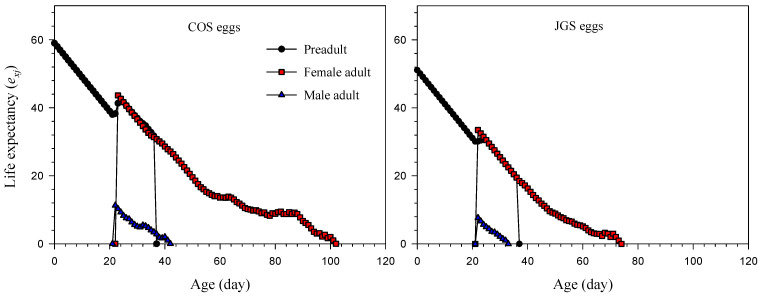
Life expectancy (*e_xj_*) of *Anastatus japonicus* parasitizing COS and JGS eggs.

**Figure 4 insects-13-00349-f004:**
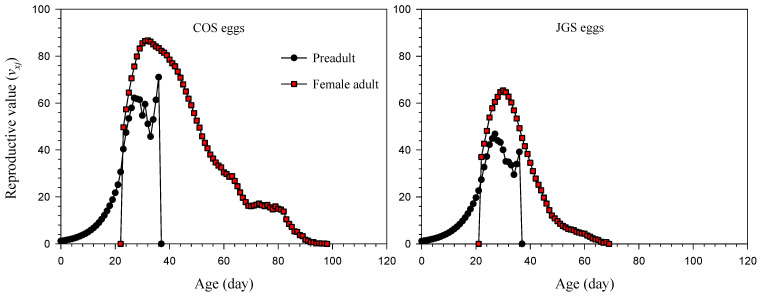
Reproductive values (*v_xj_*) of *Anastatus japonicus* parasitizing COS and JGS eggs.

**Table 1 insects-13-00349-t001:** Population cohort, adult longevity, total longevity, adult preoviposition period, total preoviposition period, oviposition days, eggs laid per oviposition day, fecundity, and age at 0.5 *l_x_* of *A. japonicus* parasitizing COS and JGS eggs.

Parameters	COS Eggs ^1^	JGS Eggs ^1^
Initial cohort size (female: male)	20:20	24:24
Matched cohort size (female: male)	259:77	175:36
Female adult longevity (days)	40.50 ± 0.84 ^a^	29.43 ± 0.54 ^b^
Female total longevity (days)	66.64 ± 0.87 ^a^	55.49 ± 0.57 ^b^
Male adult longevity (days)	10.19 ± 0.50 ^a^	6.47 ± 0.37 ^b^
Male total longevity (days)	33.27 ± 0.51 ^a^	29.69 ± 0.42 ^b^
Adult preoviposition period (days)	0.85 ± 0.03 ^a^	0.92 ± 0.03 ^a^
Total preoviposition period (days)	26.99 ± 0.14 ^a^	26.98 ± 0.19 ^a^
Oviposition days	35.88 ± 0.71 ^a^	24.02 ± 0.46 ^b^
Eggs laid per oviposition day	10.30 ± 0.05 ^a^	7.53 ± 0.06 ^b^
Total fecundity (eggs/female)	369.7 ± 7.0 ^a^	180.9 ± 3.8 ^b^
Age at 0.5 *l_x_* (days)	61.00 ± 0.85 ^a^	53.00 ± 0.89 ^b^

^1^ Values are mean ± SE, and different letters within the same row indicate significant differences in the means (paired bootstrap test, *B* = 100,000, *p* < 0.05).

**Table 2 insects-13-00349-t002:** Population parameters of *Anastatus japonicus* parasitizing COS and JGS eggs.

Population Parameter	COS Eggs ^1^	JGS Eggs ^1^
Net reproduction rate (*R*_0_) (offspring)	284.9 ± 10.0 ^a^	150.0 ± 5.6 ^b^
*R*_0, F_ (female offspring)	205.3 ± 6.9 ^a^	118.7 ± 4.3 ^b^
*R*_0, M_ (male offspring)	79.7 ± 3.4 ^a^	31.4 ± 1.8 ^b^
Intrinsic rate of increase (*r*) (day^−1^)	0.1466 ± 0.0011 ^a^	0.1419 ± 0.0012 ^b^
Finite rate of increase (*λ*) (day^−1^)	1.1579 ± 0.0012 ^a^	1.1525 ± 0.0014 ^b^
Mean generation time (*T*) (days)	38.55 ± 0.19 ^a^	35.31 ± 0.20 ^b^
Net non-effective parasitism rate (*G*_0_)	0.00 ^b^	9.60 ± 0.61 ^a^
Net parasitism rate (*P*_0_) (eggs/parasitoid)	284.9 ± 10.0 ^a^	159.6 ± 6.0 ^b^
Transformation rate (*Q_p_*)	1.00 ± 0.00 ^b^	1.0640 ± 0.0033 ^a^

^1^ Values are mean ± SE, and different letters within the same row indicate significant differences between treatments (paired bootstrap test, *B* = 100,000, *p* < 0.05).

## Data Availability

Not applicable.

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
