# Peer review of "Demography and Fitness of Anastatus japonicus Reared from Antheraea pernyi as a Biological Control Agent of Caligula japonica"

_insects, 2022, doi:10.3390/insects13040349_

Round 1

Reviewer 1 Report

The article “Demography and fitness of Anastatus japonicus reared from
Antheraea pernyi as a biological control agent of Caligula japonica

It is an interesting topic and the English is clear. Some sections of the manuscript should be better clarified.

The objective of the paper is to verify the best method of massive Parasitoid rearing in laboratory conditions, to develop a cost-effective biological control program against JGS.

As far as I am concerned, I do not believe that the method of extracting eggs from the abdomen of adult females of the replacement host and washing them in distilled water can be considered a valid breeding method for mass rearing. The method appears somewhat laborious and impractical.
However, the easier parasitization of unfertilized eggs by the replacement host represents a good biological datum to be investigated further.

In my opinion, this article can be published in this journal after some major revisions.

Please check the following parts of text:

Pag.2 line 60-62 you wrote: Anastatus bifasciatus Geoffroy have been reported as the dominant egg parasitoid species against the invasive brown marmorated stink bug Halyomorpha halys Stål in America and
 Europe [12-15].

This is not really true: A. bifasciatus resulted the predominant parasitoid of H. halys among the native parasitorids but the field data show a limited impact on H. halys egg masses (Zapponi, L.; Tortorici, F.; Anfora, G.; Bardella, S.; Bariselli, M.; Benvenuto, L.; Bernardinelli, I.;
Butturini, A.; Caruso, S.; Colla, R.; et al. Assessing the Distribution of Exotic Egg Parasitoids of Halyomorpha halys in Europe with a Large-Scale
Monitoring Program. Insects 2021, 12, 316. https://doi.org/10.3390/ insects12040316); (Stahl, J.; Babendreier, D.; Marazzi, C.; Caruso, S.; Costi, E.; Maistrello, L.; Haye, T. Can Anastatus bifasciatus Be Used for
Augmentative Biological Control of the Brown Marmorated Stink Bug in Fruit Orchards? Insects 2019, 10, 108. [CrossRef])

better define this paragraph, considering the available bibliography, please.

pag 3. line 115- please, define whether the eggs were naturally laid by COS

pag 4. line 178 - 180 - in this case (Ro) defines the average number of emerged female progeny (Birch, L. (1948). The intrinsic rate of natural increase of an insect population. The Journal of Animal Ecology, 15-26.)

pag 5. line 215-216 - please define and justify why the initial cohort size of a. japonicus was different between COS and JGS

pag 9. line in this case it is not clear how one can get to determine such an important parameter as host acceptance starting from eggs laid by two different hosts. the egg environment, with its characteristics and specificities (related to the species), are fundamental for verifying the suitability of a parasitoid. in my opinion they had to compare different eggs belonging to the same host species, in one case laid and in the other taken from the abdomen of the mature female

please, in this case, you have to consider to change this paragraph

pag 9 - line  346 - change ahd with had

pag 9 - line  347 - add not before fertilizatized

I recommend improving the discussion part by emphasizing the interesting biological and behavioral factors more than mass rearing

Author Response

The article “Demography and fitness of Anastatus japonicus reared from Antheraea pernyi as a biological control agent of Caligula japonica.

It is an interesting topic and the English is clear. Some sections of the manuscript should be better clarified.

The objective of the paper is to verify the best method of massive Parasitoid rearing in laboratory conditions, to develop a cost-effective biological control program against JGS.

As far as I am concerned, I do not believe that the method of extracting eggs from the abdomen of adult females of the replacement host and washing them in distilled water can be considered a valid breeding method for mass rearing. The method appears somewhat laborious and impractical. However, the easier parasitization of unfertilized eggs by the replacement host represents a good biological datum to be investigated further. In my opinion, this article can be published in this journal after some major revisions.

Answer: We thank the reviewer for the positive comments and suggestions; we have now included some clarifications in the revised manuscript to reflect the comments. The method of extracting eggs from the abdomen of adult females of the replacement host and washing them in distilled water has been used widely in China. COS eggs have been mass-produced and widely used as factitious hosts for rearing several egg parasitoids. This host speceis is a rich local resource and available in all seasons. It has the advantages of low production costs, easy transportation, and high reproductive efficiency. We have used this host and this method to mass-rear several Trichogramma parasitoids for field release. The processes of mass rearing of these egg parasitoids using unfertilized, extracted, and washed COS eggs have already been mechanized or automated, including collection and cleaning (drying) of host eggs, preparation of egg cards, parasitoids inoculation, and selection of parasitized host eggs (please see Zang et al. 2021. Biological control with Trichogramma in China: history, present status and perspectives. Annu. Rev. Entomol. 66: 463–484. Wang et al. 2020. Manually-extracted unfertilized eggs of Chinese oak silkworm, Antheraea pernyi, enhance mass production of Trichogramma parasitoids. Entomologia Generalis 40: 397–406).

Please check the following parts of text:

Pag.2 line 60-62 you wrote: Anastatus bifasciatus Geoffroy have been reported as the dominant egg parasitoid species against the invasive brown marmorated stink bug Halyomorpha halys Stål in America and Europe [12-15].

This is not really true: A. bifasciatus resulted the predominant parasitoid of H. halys among the native parasitorids but the field data show a limited impact on H. halys egg masses (Zapponi, L.; Tortorici, F.; Anfora, G.; Bardella, S.; Bariselli, M.; Benvenuto, L.; Bernardinelli, I.; Butturini, A.; Caruso, S.; Colla, R.; et al. Assessing the Distribution of Exotic Egg Parasitoids of Halyomorpha halys in Europe with a Large-Scale Monitoring Program. Insects. 2021, 12, 316. https://doi.org/10.3390/ insects12040316); (Stahl, J.; Babendreier, D.; Marazzi, C.; Caruso, S.; Costi, E.; Maistrello, L.; Haye, T. Can Anastatus bifasciatus Be Used for Augmentative Biological Control of the Brown Marmorated Stink Bug in Fruit Orchards? Insects 2019, 10, 108. [CrossRef]) better define this paragraph, considering the available bibliography, please.

Answer: Thank you for pointing this and for sharing useful information about the biological control of BMSB . We have corrected these statements and added the citations of these two bibliography ( See line 60-65, page 2.

Page 3. line 115- please, define whether the eggs were naturally laid by COS.

Answer: DONE. The COS eggs were unfertilized, extracted and washed. We have added a sentence to explain this in the text, see lines 118, page 3.

Page 4. line 178 - 180 - in this case (R0) defines the average number of emerged female progeny (Birch, L. (1948). The intrinsic rate of natural increase of an insect population. The Journal of Animal Ecology, 15–26.).

Answer: Yes, R0 refers to the average number of emerged female offspring, but for this study this also includeds male offspring as we used the two-sex life table method (please see Wen, M.F.; Chi, H.; Lian, Y.X.; Zheng, Y.H.; Fan, Q.H.; You, M.S. Population characteristics of Macrocheles glaber (Acari: Macrochelidae) and Stratiolaelaps scimitus (Acari: Laelapidae) reared on a mushroom fly Coboldia fuscipes (Diptera: Scatopsidae). Insect Sci. 2017, 1–11. )

Page 5. line 215-216 - please define and justify why the initial cohort size of A. japonicus was different between COS and JGS.

Answer: In fact, the initial cohort sizes of A. japonicus all were 25 on both COS and JGS eggs. Unfortunately, in four (on COS eggs) and one (on JGS eggs) replicates, the female adults escaped during the changes of fresh egg cards or accidently killed (stuck to death to the honey) eventually, we excluded these replicate and used 20 and 24 valid replicates. We have added this explanation.

Page 9. line in this case it is not clear how one can get to determine such an important parameter as host acceptance starting from eggs laid by two different hosts. the egg environment, with its characteristics and specificities (related to the species), are fundamental for verifying the suitability of a parasitoid. in my opinion they had to compare different eggs belonging to the same host species, in one case laid and in the other taken from the abdomen of the mature female. Please, in this case, you have to consider to change this paragraph.

Answer: We agree with the reviewer. However, as we cited our own studies in the introduction that the effect of differently treat eggs on the parasitoid’s host acceptance has been previously reported (please see Chen et al. 2022. Chinese oak silkworm Antherae pernyi egg, a suitable factitious host for rearing eupelmid egg parasitoids. Pest. Manag. Sci. DOI: 10.1002/ps.6796). Our main purpose of this study was to determine the performance of A. japonicus reared from this factitious host (COS) on the target host (JGS), by comparing the parasitoid’s demography and effectiveness on both hosts in order to develop a cost-effective biological control program against JGS using this parasitoid.

Page 9 – line 346 - change ahd with had.

Answer: DONE. See line 345, page 9.

Page 9 – line 347 – add not before fertilized.

Answer: DONE. See line 346, page 9.

I recommend improving the discussion part by emphasizing the interesting biological and behavioral factors more than mass rearing.

Answer: Thanks for your thoughtful suggestion. We have already provided considerable discussions on potential behavioral and physiological mechanisms that could have affected the different demography and parasitic effectiveness of the parasitoid on the two hosts. We discussed possible effect of quality and host egg surface characteristics and natal host (rearing experience) on the parasitoid’s performance (host location, host acceptance and suitability). . See lines 299-322, page 9.

Reviewer 2 Report

The presented study shows standard measurements of life tables of one parasitoid species reared in lab conditions on two hosts. There is little scientific importance but potentially high practical use. 

The manuscript requires multiple minor improvements in presenting results, such as: 

Throughout: unify font size.

Abstract, l. 27: add "reared for six generations on COS eggs"

l.32: values for JGS missing

Methods, l. 196: Delete repeated equation.

Results: Do not repeat values in the text if they are presented in the tables just below it. Anyway, some values are given only for one host species.

Abbreviations like APOP and TPOP in text and tables are useless because authors always repeat the full term.

Table 1: What are the values after +-?

Lines 234+: On the contrary, the values presented in this paragraph cannot be seen in the graph below and their origin is doubtful. It looks like if some values were simply estimated by eye from the graph and are not exact. Such as "stable from age 45 to 52 d - I see to 54 days; and "decreased sharply after age 37 d" - I would say after 43 days. But there is no test to identify the borders between stagnation and decrease. 

l. 248: repeated JGS.

l. 248: the differences are not visible during this initial increase period and if existing, they would not be important.

l. 258: The "peak" is not correct term if it is an initial value of general decrease. 

l. 283: There is no test to compare Qp. 

Table 2: Some variables, at least G0 are not comparable between hosts - fertilized developing and unfertilized eggs without real defense activity. 

Author Response

The presented study shows standard measurements of life tables of one parasitoid species reared in lab conditions on two hosts. There is little scientific importance but potentially high practical use.

Answer: We thank the reviewer for the positive comments and suggestions; we have now included some clarifications in the revised manuscript to reflect the comments.

The manuscript requires multiple minor improvements in presenting results, such as: Throughout: unify font size.

Answer: DONE.

Abstract, l. 27: add "reared for six generations on COS eggs"

Answer: Thanks for your suggestion. However, we are not sure if it is necessary to emphasis this information in the Abstract. In this study, we did not test specifically possible effect of rearing generation on the parasitoid’s ftiness and it just happened that the wasps we used in this study had been reared for six generations on COS eggs. We are afraid that readers may easily relate this study to the effect of rearing generation if we emphasized this information in the Abstract. l.32: values for JGS missing. Answer: DONE. See line 33.

Methods, l. 196: Delete repeated equation.

Answer: DONE. See line 196, page 5.

Results: Do not repeat values in the text if they are presented in the tables just below it. Anyway, some values are given only for one host species.

Answer: DONE. See line 222, page 6 and line 283-283, page 8. We’ve removed all these values in the text as they are already presented in the Tables.

Abbreviations like APOP and TPOP in text and tables are useless because authors always repeat the full term.

Answer: DONE. We have deleted these (and some others) abbreviations that seem to be not necessary throughout the text.

Table 1: What are the values after +-?

Answer: The values after +- were referred to means ± SE. We have added this information note for Table 1, see page 6.

Lines 234+: On the contrary, the values presented in this paragraph cannot be seen in the graph below and their origin is doubtful. It looks like if some values were simply estimated by eye from the graph and are not exact. Such as "stable from age 45 to 52 d - I see to 54 days; and "decreased sharply after age 37 d" - I would say after 43 days. But there is no test to identify the borders between stagnation and decrease. 

Answer: We have double-checked our data to ensure the correctness of these values in this paragraph. Those values were not roughly estimated by eye from the graph, rather they were estimated by TWOSEX-MSChart program.

  1. 248: repeated JGS.

Answer: DONE. See line 247, page 7.

  1. 248: the differences are not visible during this initial increase period and if existing, they would not be important.

Answer: DONE. We have deleted this sentence.

  1. 258: The "peak" is not correct term if it is an initial value of general decrease.

Answer: DONE. We have replaced “peak” with “highest ”, see line 257, page 7.

  1. 283: There is no test to compare Qp.

Answer: DONE. We have added this information, see Table 2.

Table 2: Some variables, at least G0 are not comparable between hosts - fertilized developing and unfertilized eggs without real defense activity.

Answer: G0 refers to the net non-effective parasitism rate (i.e., the number of host eggs that were parasitized successfully, but no parasitoid offspring emerged) and is usually used as a parameter to reflect the fitness or adaptability of the parasitoid to the host. In our case, we intended to use G0 to compare the relative adaptability of A. japonicus on these two different hosts.

Round 2

Reviewer 1 Report

I thank the authors for the improvements made to the text and for the explanations provided.
There remains only one point that I did not notice during the first reading and that needs correction.

Page 2. Line 68 A. bifasciatus, to the best of my knowledge is not a predominant parasitoid in America but only in Europe. If not, please enter further bibliography to confirm. thank you

Author Response

COMMENTS FOR AUTHORS

I thank the authors for the improvements made to the text and for the explanations provided. There remains only one point that I did not notice during the first reading and that needs correction.

Page 2. Line 68 A. bifasciatus, to the best of my knowledge is not a predominant parasitoid in America but only in Europe. If not, please enter further bibliography to confirm. thank you.

Answer: DONE. We thank you for the value suggestion again. We have re-checked the point and agreed with your opinion. We have deleted "in America" in text, see line 66, page 2.